# The Identification of ECG Signals Using WT-UKF and IPSO-SVM

**DOI:** 10.3390/s22051962

**Published:** 2022-03-02

**Authors:** Ning Li, Longhui Zhu, Wentao Ma, Yelin Wang, Fuxing He, Aixiang Zheng, Xiaoping Zhang

**Affiliations:** 1School of Electrical Engineering, Xi’an University of Technology, Xi’an 710048, China; 2190321214@stu.xaut.edu.cn (L.Z.); mawt@xaut.edu.cn (W.M.); 3160411041@stu.xaut.edu.cn (Y.W.); 2180320030@stu.xaut.edu.cn (F.H.); 2School of Humanities and Foreign Languages, Xi’an University of Technology, Xi’an 710048, China; zax@xaut.edu.cn; 3Department of Electronic, Electrical, and Systems Engineering, School of Engineering, University of Birmingham, Birmingham B15 2TT, UK; x.p.zhang@bham.ac.uk

**Keywords:** electrocardiogram identification, wavelet transform, unscented Kalman filter, parameter optimization, improved particle swarm optimization, support vector machine

## Abstract

The biometric identification method is a current research hotspot in the pattern recognition field. Due to the advantages of electrocardiogram (ECG) signals, which are difficult to replicate and easy to obtain, ECG-based identity identification has become a new direction in biometric recognition research. In order to improve the accuracy of ECG signal identification, this paper proposes an ECG identification method based on a multi-scale wavelet transform combined with the unscented Kalman filter (WT-UKF) algorithm and the improved particle swarm optimization-support vector machine (IPSO-SVM). First, the WT-UKF algorithm can effectively eliminate the noise components and preserve the characteristics of ECG signals when denoising the ECG data. Then, the wavelet positioning method is used to detect the feature points of the denoised signals, and the obtained feature points are combined with multiple feature vectors to characterize the ECG signals, thus reducing the data dimension in identity identification. Finally, SVM is used for ECG signal identification, and the improved particle swarm optimization (IPSO) algorithm is used for parameter optimization in SVM. According to the analysis of simulation experiments, compared with the traditional WT denoising, the WT-UKF method proposed in this paper improves the accuracy of feature point detection and increases the final recognition rate by 1.5%. The highest recognition accuracy of a single individual in the entire ECG identification system achieves 100%, and the average recognition accuracy can reach 95.17%.

## 1. Introduction

With the development of information technology and the rapid popularization of the internet, biometrics has demonstrated a unique advantage in identification technology. Conventional biometric identification technology includes the following: fingerprint recognition, facial recognition, voice recognition, and other physiological feature recognition technology [1,2]. Although these types have a high recognition rate, they also have some disadvantages: they are easy to replicate and forge, they are sensitive to the recognition environment, etc. [3,4].

In recent years, electrocardiogram (ECG) signal identification has received more attention. ECG signals are electric signals generated by the human heart beat; these signals have the advantage of being difficult to forge or lose. With the development of ECG data acquisition technology, portable signal acquisition devices, such as sports watches, have been designed as highly intelligent and convenient products. Therefore, identification technology based on ECG signals has a strong theoretical research value and bright market application prospects [5,6].

Identification technology based on ECG signals has two core contents: ECG signal preprocessing and identity identification. The preprocessing of ECG signals involves obtaining feature data for identification; the identity identification involves classifying the preprocessed data based on the classification algorithm.

The preprocessing of ECG signals can be divided into filtering processing and feature detection. ECG signals initially acquired from the human body contain substantial noise. Therefore, it is necessary to filter ECG signals first. ECG signals are typical low-frequency signals. The basic denoising methods include Savitzky–Golay [7], wavelet denoising [8] and empirical mode decomposition (EMD) [9]. For a better filtering effect, S. Boda and L. El Bouny [10,11] used EMD decomposition to obtain IMF components, in which empirical wavelet transform and denoising processing were performed based on high-order statistics. However, as the EMD algorithm itself lacks a complete theoretical basis, the phenomenon of modal confusion is prone to occur; thus, the denoising effect cannot be guaranteed in the ECG denoising. D. Zhang [12] used wavelet energy to select wavelet coefficients that required threshold denoising, and they used a sub-band smoothing filter to further denoise ECG signals. This method solved the problem of signal distortion caused by the improper selection of wavelet coefficients during wavelet threshold denoising, but did not solve the problem of the choice of threshold value. Z. Wang [13] proposed a wavelet denoising method for ECG signals. This method constructs a new wavelet base to process ECG signals and uses conventional threshold methods to denoise. In this way, the features of P and T waves are preserved, but the waveform is partially distorted and the baseline wander is not removed. X. K. Wan [14] combined wavelet transform and mathematical morphology filtering to effectively suppress the limit drift. Z. Jin [15] proposed a sparse ECG signal denoising method that combined low-pass filtering and sparse recovery, but it was difficult to select and determine sparse dictionaries, and different sparse dictionaries, have a great impact on the results. However, the above denoising methods obtain good SNR, but the local characteristics of the signals are rarely preserved, making ECG feature detection and identity identification more difficult. HR.C [16] proposed the use of the root mean square error (RMSE) to denoise ECG signals. Compared with the traditional method, based on the percentage root mean square difference (PRD), this method better controlled the reconstruction of signals without reducing the compression ratio, and the reconstructed signals were not distorted. Later, HR.C [17] proposed an ECG compression denoising method based on the trigonometric function coefficient dictionary. This method can efficiently complete the compression and denoising processing of ECG signals, and the subsequent QRS wave detection accuracy achieved a value as high as 99.55%. However, neither method is effective at identity identification.

Feature detection of the filtered ECG signals can be classified into fiducial point detection and non-fiducial point detection [18,19]. In fiducial point detection, ECG signals are defined by fixed fiducial points (such as the peaks of P, QRS complex, and T waves). The time interval between these fiducial points and the amplitude at these points are defined as ECG signal characteristics [20,21]. Non-fiducial point detection does not require the detection of the fiducial point, and methods such as the reconstructed phase space [22], the principal component analysis method [23], and sparse representation [24] are usually used. However, the recognition scale of the non-fiducial point method is small, as the required algorithm is more sophisticated and the amount of extracted feature data are larger.

Commonly used methods of ECG identification include linear classifier [25] and random forest [26]. These methods may not be robust enough in practice, and the identification effect depends heavily on signal quality. Therefore, a neural network [27] is introduced for classification. U. R. Acharya [28] used a 9-layer convolutional neural network (CNN) to achieve identification on the MIT-BIH database. O. Yildirim [29] used a 16-layer CNN model for identification on the same database. The accuracy of the above two methods reached 94.03% and 95.2%, respectively. S. Chauhan [30] used the long–short-term memory (LSTM) networks to classify normal and abnormal heartbeats, with an accuracy rate of 96.45%. S. L. Oh [31] combined CNN with LSTM, using a new hybrid structure for identification, and achieved good results. D. Belo [32] used the temporal convolutional neural network (TCNN) and recurrent neural network (RNN) for identity identification, and the accuracy of these two groups of data are close to 96% and 90%. Although the neural network method has high identification accuracy, the algorithm is unable to explain its own reasoning process and reasoning basis. To obtain higher identification accuracy, a larger amount of data and deeper network structure are required.

This paper proposes an ECG signal identification method based on multi-scale wavelet transform combined with the unscented Kalman filter algorithm (WT-UKF) and the improved particle swarm optimization support vector machine (IPSO-SVM). The following methods are used in the preprocessing of the ECG data: first, the WT-UKF method is used to filter ECG signals. While eliminating the noise interference, it greatly preserves the local characteristics of the ECG signals. Then, the dyadic spline wavelet transform is used to extract the feature points of QRS complex waves, and windowed processing is applied to eliminate the interference of the peak points of R waves, and extract feature points of P and T waves. Finally, composed of the obtained feature points, 22 feature vectors characterize the ECG signals and reduce the ECG data dimension for identification. In ECG signal identification, this paper uses the support vector machine (SVM) method, which is more effective on the small sample training set, and uses the IPSO algorithm to optimize the parameters of SVM, allowing the SVM classifier to determine the parameters independently and prevent the occurrence of the local maximum, making the classification as accurate as possible. The system flow diagram of this paper is shown in Figure 1.

The innovation of this paper is mainly reflected in the following two aspects:In order to preserve the local characteristic information of ECG signals, this paper proposes the WT-UKF algorithm to filter ECG signals, which preserves the characteristics of the signals under the premise of a good filtering effect, and improves the feature point extraction of P waves and T waves.In order to ensure the accuracy of SVM in ECG identification, this paper uses the IPSO-SVM algorithm to identify ECG signals; it can adaptively select penalty parameters and hyperparameters.

The rest of this paper is arranged as follows: in Section 2, the basic algorithm principles used in this paper are introduced. In Section 3, the WT-UKF algorithm for preprocessing and the IPSO-SVM algorithm for identification are introduced. In Section 4, the MIT-BIH arrhythmia database is used for simulation, and the effectiveness and robustness of this method are verified by comparisons with other methods. Finally, Section 5 presents the results and conclusion.

## 2. Required Tools

This section briefly introduces the application of the multi-scale wavelet transform and the unscented Kalman filter algorithm in ECG signal processing. Meanwhile, it introduces the application of the SVM algorithm in ECG identification, making preparations for the construction of the ECG identification system in Section 3.

### 2.1. Multi-Scale Wavelet Transform Filtering

Clinically, the characteristics of ECG signals can mainly be found in the amplitudes of different peaks and intervals of P waves, QRS complex waves, and T waves. However, the noise in ECG signals directly affects the detection accuracy of P, QRS, and T waves, and the features of P, QRS, and T waves exist in different frequency ranges. Therefore, ECG signal preprocessing needs to be carried out simultaneously in frequency and time domains. Wavelet transform (WT) is a time–frequency localization analysis method with a fixed window area, but a variable time window and frequency window, and it has self-adaptability to signals. Multi-scale decomposition and reconstruction based on the Mallat algorithm is particularly suitable for the multi-scale filtering of ECG signals [8].

After multi-scale decomposition, ECG signals have different time and frequency resolutions on different scales. Signals with high time resolution contain more detailed information and high-frequency information, while signals with low time resolution contain more global information. The approximate coefficients and detail coefficients are reconstructed after performing the threshold processing. The reconstructed ECG signals are the signals after multi-scale filtering. Figure 2 is a diagram of the wavelet decomposition and reconstruction based on the Mallat algorithm.

In Figure 2, f(t) is the original ECG signals; *t* is the number of the time serials; j=1,2,…,J is the decomposition scale; *H* and *G* are the low- and high-pass filters in the time domain, respectively; $\;A_{j}\left[f\left(t\right)\right]$ is the approximation coefficients of wavelet decomposition after low-pass filtering on scale *j*; $\;D_{j}\left[f\left(t\right)\right]$ is the detail coefficients of wavelet decomposition after high-pass filtering on scale *j*; $\;H_{r}$ and $\;G_{r}$ are the low- and high-pass wavelet reconstruction filters, respectively, in the time domain.Taking the ECG signals of a subject as an example, the filtering effect is shown in Figure 3.

### 2.2. Unscented Kalman Filter

Although ECG signals are periodic signals, there is some noise interference, such as power line interference and motion artifacts. Additionally, the Kalman filter (KF) can effectively carry out real-time optimal estimation from the measurement data with noise, and achieve filtering processing. Therefore, KF can ensure the elimination of interference while preserving the local characteristics of the ECG signals.

The traditional KF linear filtering framework is not suitable for typical nonlinear data such as ECG signals, while the unscented Kalman filter [33] (UKF) algorithm uses unscented transformation to deal with the nonlinear transfer problem of mean and covariance, which avoids the error caused by the linearization process. Meanwhile, the conventional KF linear filtering framework was applied to achieve filtering processing. Taking the ECG signals of a subject as an example, the filtering effect is shown in Figure 4.

### 2.3. PSO-SVM Algorithm

The support vector machine (SVM) is a supervised learning algorithm, which is essentially a two-classifier algorithm. SVM uses the principle of structural risk minimization (SRM) to complete the classification operation. Compared with the principle of empirical risk minimization (ERM) adopted by the artificial neural network, SRM has stronger interpretability and better generalization ability, so SVM is widely used in the field of ECG classification.

At the same time, in order to achieve the multi-classification effect of ECG identification, this paper adopted the 1-1 (one-versus-one, OVO) discrimination method to achieve SVM multi-classification. The basic concept of the OVO discrimination method to complete the multi-class recognition of ECG signals is as follows: assuming that the training set of ECG signal samples contains k>2 category data, select two different ECG signal categories to form a sub-classifier of SVM and, therefore, there are k(k−1)/2 SVM sub-classifiers in k categories. When constructing an SVM sub-classifier of category A and category B, the sample data of category A and category B in the ECG signal sample data set were used as training data. In the multi-category test, the test data on k(k−1)/2 SVM sub-classifiers were tested, respectively. If it belongs to a certain category, its score will “plus one”, accumulate the scores of each category, and select the category with the highest score as the category of the test data.

ECG identification is a nonlinear classification problem, which needs to select an appropriate kernel function to complete the mapping of high-dimensional space [34]. Additionally, when using SVM, the optimal free parameters are selected to optimize the classification effect of SVM, such as the selection of the penalty parameter (*c*) and hyper parameter (*g*). Usually, the optimal parameters are obtained according to experience. In [35], the PSO algorithm is used to complete the adaptive selection of the penalty parameter (*c*) and the hyper parameter (*g*) in the SVM algorithm, and the accuracy can reach 89.72% in ECG signal classification. However, due to the defects of the PSO algorithm, the SVM classification effect is not optimal.

## 3. Proposed Method in ECG Identification System

### 3.1. Data Preprocessing Algorithm Based On WT-UKF

Considering the contents of Section 2.1 and Section 2.2, this paper proposes a data preprocessing algorithm based on the decomposition and reconstruction of conventional multi-scale wavelet transform combined with the unscented Kalman filter algorithm (WT-UKF). This algorithm applies the UKF algorithm to multi-scale analysis, replacing the conventional threshold function, thus avoiding the defect of losing data characteristics in the traditional wavelet threshold denoising process. The basic flow of the algorithm is shown in Figure 5.

Using the MIT-BIH database as an example, the sampling frequency of the data used in this paper was 360 Hz [32]. According to the theory of WT, the frequency range of the approximate coefficients of the first scale is 0–90 Hz, and that of the second scale was 0–45 Hz. The frequency range of the detail coefficients of the first scale was 90–180 Hz, and that of the second scale was 45–90 Hz, etc. The frequency range of the approximate coefficients of the eighth scale was 0–0.781 Hz, and that of the detail coefficients of the eighth scale was 0.781–1.563 Hz.

The extraction of the ECG signal feature point is mainly affected by power line interference and baseline wander interference. Power line interference energy is mainly concentrated at about 60 or 50 Hz, and baseline wander energy is mainly concentrated in the 0–1 Hz range [10].

According to the flow in Figure 5, the decomposition of ECG signals f(t) was completed at eight scales based on Formula (1), and eight approximate coefficients $A_{j}\left[f\left(t\right)\right]$ and eight detail coefficients $D_{j}\left[f\left(t\right)\right]$ were obtained, where *j* = 1–8. Ten samples of subjects were selected in the database, and three different wavelet basis functions of Sym8, db5, and db8 were used for ten filtering tests. Finally, the db8 wavelet was selected as the mother wavelet in this paper.

Through the above analysis, it can be concluded that the baseline wander in the ECG signal mainly existed in the approximate coefficients of the 8th scale; therefore, in order to eliminate the baseline wander, this paper set $A_{8}\left[f\left(t\right)\right]$ to 0; the high-frequency noise in ECG signals mainly existed in detail coefficients of the first and second scale. As the first scale of detail coefficients contained most of the high-frequency noise, this paper set $D_{1}\left[f\left(t\right)\right]$ to 0 to eliminate most of the high-frequency noise interference, and then carried out UKF processing on $D_{2}\left[f\left(t\right)\right]$ to obtain $D_{2}^{U}\left[f\left(t\right)\right]$, which eliminated power line interference and high-frequency noise while ensuring that the characteristics of original signals of the second scale were preserved as much as possible. After finishing the above-mentioned processing, the processed detail coefficients and approximate coefficients were used to complete the wavelet reconstruction based on Section 2.1, and the denoised ECG signals were obtained. The reconstruction formula and conditions are shown in Formula (1), where $A_{0}^{U}\left[f\left(t\right)\right]$ denotes the filtered ECG signals.
(1)$$\left\{\begin{array}{c} A_{7}\left[f\left(t\right)\right]=2\left\{\sum_{k}g\left(t-2k\right)D_{8}\left[f\left(k\right)\right]\right\}\\ \vdots \\ A_{1}^{U}\left[f\left(t\right)\right]=2\left\{\sum_{k}h\left(t-2k\right)A_{2}\left[f\left(k\right)\right]+\sum_{k}g\left(t-2k\right)D_{2}^{U}\left[f\left(k\right)\right]\right\}\\ A_{0}^{U}\left[f\left(t\right)\right]=2\left\{\sum_{k}h\left(t-2k\right)A_{1}^{U}\left[f\left(k\right)\right]\right\} \end{array}\right.$$

In order to measure the effectiveness of the denoising method, the objective functions selected in this paper were the signal-to-noise ratio (SNR) and root mean square error (RMSE). The formulas of the two objective functions are as follows:(2)$$\mathrm{SNR}=20\times \mathrm{lo}g_{10}\left(A_{signal}/A_{noise}\right)$$
where $A_{signal}$ is the signal amplitude and $A_{noise}$ is the noise amplitude.
(3)$$\mathrm{RMSE}=\sqrt{\frac{1}{m}\sum_{o=1}^{m}{\left(y_{o}-{\hat{y}}_{o}\right)}^{2}}$$
where *m* is the signal length, $y_{o}$ is the useful signals without noise, and ${{\hat{y}}_{o}}^{2}$ is the filtered ECG signals after denoising.

### 3.2. ECG Identity Recognition Algorithm Based On IPSO-SVM

In this paper, an ECG identification algorithm based on IPSO-SVM is proposed, which solves the problem of SVM parameter selection and achieves the optimal classification effect.

#### 3.2.1. SVM Kernel Function Selection

According to Section 2.3, in order to make the SVM classification model of ECG achieve the maximum accuracy of nonlinear classification, it is necessary to select a suitable kernel function to complete the high-dimensional space reflection. In order to select the suitable kernel function to construct the SVM classifier, this paper applied a statistical method for verification, called m-fold cross validation [36]. A 3-fold cross validation was performed on the entire ECG feature sets, including the training set and the testing set. The overall accuracy estimates of different kernel functions are shown in Table 1.

According to the results in Table 1, this paper used the RBF kernel function to construct the classifier model, and the kernel function is defined as follows:(4)$$K\left(x,y\right)=e^{-g{∥x-y∥}^{2}}$$
where x and y are samples or vectors, respectively; *g* is the only hyper parameter of the Gaussian kernel function; and $∥x-y∥$ is the norm of the vector. The result of this formula is a specific value, which represents the relationship between the vectors. It can be seen from the formula that the selection of parameter *g* has a great influence on the accuracy of the classifier and, thus, the value of *g* is one of the optimization goals.

According to Section 2.3, the value of the penalty parameter (*c*) that needs to be set before the SVM training model was used. The penalty parameter was used to indicate the degree of emphasis on individual points. The larger the value of *c* is, the more attention is paid to those outliers; the smaller the value of *c* is, the less attention is paid to those outliers. If the penalty parameter tends to infinity, samples with classification errors are not allowed to exist, which will cause overfitting problems; when *c* tends to 0, no attention is paid to whether the classification is correct, it is not possible to obtain a meaningful solution, and the algorithm does not converge. Only when the penalty parameter is selected appropriately can the SVM training model achieve the maximum classification accuracy. Therefore, the value of *c* is another optimization goal.

#### 3.2.2. IPSO Algorithm

The basic concept of the PSO algorithm is to seek the optimal solution through collaboration and information sharing among individuals in the population. PSO is initialized as a population of random particles (random solution), and then the optimal solution is found through iteration. In each iteration, the particle updates itself by tracking the individual optimal and the population optimal (ibest, pbest). After finding these two optimal values, the particle uses the following formula to update its speed and position:(5)$$\left\{\begin{array}{c} V_{i+1}=\omega_{v}\times V_{i}+f\left(x_{i},ibest_{i}\right)+f\left(x_{i},pbest_{i}\right)\\ f\left(x_{i},ibest_{i}\right)=C_{1}\times rand\left(\right)\times \left(ibest_{i}-x_{i}\right)\\ f\left(x_{i},pbest_{i}\right)=C_{2}\times rand\left(\right)\times \left(pbest_{i}-x_{i}\right)\\ x_{i+1}=x_{i}+V_{i} \end{array}\right.$$
where $x_{i}$ is the particle’s own position, $V_{i}$ is the particle’s own velocity, $C_{1}$ is its own learning factor, $C_{2}$ is the social learning factor, $\omega_{v}$ is the velocity elasticity factor, rand() is a random number between 0 and 1, and the maximum value of $V_{i}$ is $V_{\max}$. When $V_{i}$ update exceeds $V_{\max}$, take $V_{i}\;=\;V_{\max}$, all *i* represents the population size.

In order to increase the convergence speed of the IPSO algorithm, based on PSO algorithm, this paper took $\omega_{v}$ in Formula (3) as a value that changes with the number of iterations, and the value of $\omega_{v}$ is as follows:(6)$$\omega_{v}^{t}=\frac{\left(\omega_{start}-\omega_{end}\right)\left(I_{\max}-I\right)}{I_{\max}}+\omega_{end}$$
where $\omega_{v}^{t}$ is the value of $\omega_{v}$ under the current iteration number, $\omega_{start}$ is the initial velocity elasticity factor, $\omega_{end}$ is the velocity elasticity factor when the maximum iteration number is reached, *I* is the current iteration number, and $I_{\max}$ is the maximum iteration number.

In order to adaptively adjust the global search range, $\;C_{1}$ and $\;C_{2}$ in Formula (3) were taken as the values that change with the number of iterations, and the values of $\;C_{1}$ and $\;C_{2}$ are as follows:(7)$$C_{1,2}={C_{\max}}_{\left(1,2\right)}+\left({C_{\min}}_{\left(1,2\right)}-{C_{\max(}}_{1,2)}\right)\left(I/I_{\max}\right)$$
where $\;C_{1,2}$ is the value of $C_{1,2}$ under the current iteration number, ${C_{\max}}_{\left(1,2\right)}$ is the maximum learning factor value of $C_{1}$ and $C_{2}$, and ${C_{\min}}_{\left(1,2\right)}$ is the minimum learning factor value of $C_{1}$ and $C_{2}$. *I* and $I_{\max}$ are the same as (4).

In order to avoid the algorithm falling into the local optimal solution during dual-parameter optimization, the IPSO algorithm used in this paper introduced an adaptive mutation step before each fitness calculation of the traditional PSO algorithm. First, randomly determine whether mutation is required; if required, then determine whether the parameter that needs to be mutated is c or g, and perform the final mutation according to Equation (Equation 6).
(8)$$P=\left(P_{\max}-P_{\min}\right)\times rand\left(\right)+P_{\min}$$
where *P* is the parameter that needs to be mutated; $P_{\max}$ and $P_{\min}$ are the maximum and minimum values of the parameter, which are random numbers between 0 and 1, respectively.

#### 3.2.3. IPSO-SVM Algorithm

After ECG signal preprocessing is completed, the steps of IPSO-SVM algorithm proposed in this paper are as follows:

Step 1. Input all ECG signal feature vectors obtained, initialize the parameters *c* and *g*, and build multiple-classifiers for ECG signal classification and training.

Step 2. Calculate the fitness values corresponding to different parameters *c* and *g* through the fitness function, and determine *c* and *g* corresponding to the individual optimal fitness and the group optimal fitness.

Step 3. Update and iterate *c* and *g* according to formulas (5)–(8); then, update the individual optimal and population optimal fitness and their corresponding *c* and *g*, respectively. After the termination conditions are met, obtain the iteration ends and the optimal *c* and *g*.

Step 4. Build multiple classifiers according to the obtained optimal *c* and *g*, realize ECG signal identification and output the best ECG identification accuracy rate.

The block diagram of the IPSO-SVM algorithm used in this paper is shown in Figure 6.

In the IPSO algorithm in this paper, in terms of Equations (6) and (7), various parameters were tested many times; the maximum number of iterations was 50, the population size was 20, $\omega_{start}$ was 0.9, $\omega_{end}$ was 0.4, ${C_{\max}}_{\left(1,2\right)}$ is 2, and ${C_{\min}}_{\left(1,2\right)}$ was 0.5. The value of *c* was initialized to 10 and the value of *g* to 2. In summary, the SVM classification accuracy was selected as the fitness function.

## 4. Simulation Verification

The original ECG data used in this paper were obtained from the (MIT-BIH) arrhythmia ECG database and MIT-BIH normal sinus rhythm database [37,38]. The BIH database of the Massachusetts Institute of Technology is one of the three most recognized ECG databases in the world. The MIT-BIH arrhythmia database contains extracts of 48 half-hour two-channel dynamic electrocardiogram records from 47 subjects in the arrhythmia laboratory study. In order to ensure the practicability of the algorithm, the data of 16 subjects whose ECG signals were greatly different from the standard ECG signals in the database were removed, and the duplicate data of a subject were removed at the same time. This paper used the method in Section 3 to process the data of 30 subjects, with 150 ECG signal heartbeat cycles extracted for each subject; thus, there were 4500 sets of data in total. The MIT-BIH normal sinus rhythm database contains the ECG records of 18 subjects. This paper used the method in Section 3 to process the data of 10 subjects, and 150 ECG signal cycles were extracted for each subject, so there were 1500 sets of data in total.

### 4.1. Improvement in the Effect Verification of Multi-Scale Wavelet Transform

Under the (MIT-BIH) arrhythmia database, the comparison of the effect of the WT-UKF algorithm on the existing conventional wavelet threshold denoising is shown in Table 2. SNR and RMSE are the signal-to-noise ratio and root mean square error, respectively. Suppose the signals before denoising are the ECG signals of a subject superimposed with 0.3 Hz and 50 Hz sinusoidal signals.

It can be seen from Table 2 that the method proposed in this paper had a higher signal-to-noise ratio after denoising and a smaller root mean square error; thus, the overall denoising effect was better. Moreover, due to the time-varying ECG signals, this method can improve the extraction rate of P and T waves, and the comparison of filtering effects is shown in Figure 7 and Figure 8.

After using the WT-UKF algorithm to complete the above-mentioned ECG data denoising, this paper applied the feature point detection method based on the dyadic spline wavelet transform in [39] to locate the QRS complex. At the same time, windowing processing was used to extend this method to the location of P and T waves, so as to extract the sampling time points and amplitudes of QRS, P and T waves.

Figure 7 and Figure 8 demonstrate the feature extraction results of the QRS complex and P waves, respectively, in the ECG signals of subject 123. It can be seen from the figures that the QRS complex could be extracted more accurately by using the WT-UKF denoising method proposed in this paper. Compared with conventional wavelet threshold denoising, the WT-UKF denoising method improved the extraction results of P-wave features and avoided the wrong extraction of P-wave features in stationary periods. In terms of T-wave extraction, WT and WT-UKF denoising methods both had better results in the T-wave feature extraction of subject 123, and thereby, subject 103 was selected for comparison. As shown in Figure 9, it can be seen that T-wave feature extraction is effective. The problem that the starting point of T waves appeared in the QRS complex was avoided.

### 4.2. ECG Identification Based On Support Vector Machine

After obtaining the features of QRS, P, and T waves in the time domain in Section 4.1, each ECG cycle took 16 distance feature vectors: R-R, R-Q, R-S, R-P, R-T, R-P${}_{Begin}$, R-P${}_{End}$, R-T${}_{Begin}$, R-T${}_{End}$, Q-P, Q-P${}_{Begin}$, S-T, S-T${}_{End}$, P-T, P${}_{Begin}$-P${}_{End}$, T${}_{Begin}$-T${}_{End}$, and 6 amplitude feature vectors: R-Q, R-S, Q-P, S-T, P${}_{Begin}$-P, T${}_{Begin}$-T. These 22 feature vectors were used to characterize an ECG cycle signal, so as to achieve the effect of dimensionality reduction and increase the speed of the algorithm. Finally, 70% cycles were randomly extracted from the obtained feature set as the training set, and the remaining 30% cycles were used as the testing set.

#### 4.2.1. The Value of *c* and *g*

The parameters *c* and *g* in the conventional SVM are usually set according to the human experience. In order to obtain a better classification accuracy, this paper took the value of *c*, respectively, as 2, 2${}^{2}$, 2${}^{4}$, 2${}^{6}$, 2${}^{8}$, 2${}^{10}$, 2${}^{11}$, and the default value of *g* as value 1. The influence of different c values on the recognition accuracy was observed by performing three-fold cross validation on the training set data of 30 subjects. The results are shown in Table 3.

It can be seen from Table 3 that *c* had a higher accuracy rate at 2${}^{8}$, 2${}^{10}$ and 2${}^{11}$; therefore, under the values of these three penalty parameters, the influence of the hyperparameters on the accuracy was analyzed separately. Similarly, depending on human experience, the value of g was taken as $2^{-4}$, $2^{-3}$, $2^{-1}$, $2^{0}$, $2^{1}$, $2^{2}$, $2^{3}$, and the influence of different *g* values on the recognition accuracy was observed through three-fold cross-validation. The results are shown in Table 4.

According to the analysis of Table 3 and Table 4, although the highest accuracy rate appeared at $\;g=2^{-1}$ when $c=2^{8},2^{10},2^{11}$, the accuracy rate between $\;g=2^{0}$ and $\;g=2^{-1}$ did not increase continuously when $c=2^{11}$, the accuracy rate at $g=2^{2}$ was the same when $c=2^{8},2^{10},2^{11}$, and the average accuracy rate was not different. Therefore, it was found that in order to achieve the highest recognition accuracy, the values of *c* and *g* are not a fixed value, and the two parameters affect each other.

#### 4.2.2. Identification Results

The IPSO algorithm needs to set the appropriate value range of the two parameters in advance which is used for data initialization, boundary judgment and adaptive mutation. Through Table 3 and Table 4, in order to achieve the highest accuracy of the parameter values under mutual influence, $c=2^{2}$−$2^{11}$ and $g=2^{-4}$−$2^{3}$ were taken as the value ranges of the parameters in the IPSO algorithm, respectively. After determining the value range of *c* and *g*, based on the previous simulation, the testing set was used to verify the method, and the method in Section 3.2 was used to optimize the parameters. Finally, the comparison of the recognition accuracy of different methods is shown in Table 5.

The four methods of decision tree, random forest, Bayes, and logistic in Table 5 are verified by HOON KO [40]. This study improves the recognition accuracy by using the adjusted (Qi(*)Si) algorithm. The TCNN-RNN method in Table 5 was proposed by Belo D in [32]. This method uses TCNN and RNN at the same time and obtained 96% recognition accuracy in the MIT-BIH database. The CNN model in Table 5 is proposed by O. Yildirim in [29]. The model is an improved 16-layer 1D-CNN, and 95.2% accuracy was obtained under MIT-BIH data. The LSTM model in Table 5 was proposed by S. Chauhan in [30], and 96.45% accuracy was obtained in the MIT-BIH database. The PNN model in Table 5 selects parameters through human experience and obtained 94.48% accuracy in the MIT-BIH database.

In order to carry out comparative analysis from two aspects of filtering and parameter selection, this paper established a total of 5 classification models, SVM-3 (WT), SVM-1 (WT-UKF), SVM-2 (WT-UKF), SVM-3 (WT-UKF), and IPSO-SVM (WT-UKF). The functions of these models are as follows: In order to verify the influence of feature detection on recognition accuracy, SVM-3 (WT) and SVM -3 (WT-UKF) models were established in this paper. The SVM-3 (WT) model uses the traditional WT algorithm for filtering, and the SVM-3 (WT-UKF) model uses the WT-UKF algorithm for filtering. These two models are used in [39] for feature detection to obtain the feature data set, and $c=2^{10}$, $g=2^{-1}$ according to Table 3 and Table 4. In order to verify the influence of parameters *c* and *g* on recognition accuracy, SVM-1 (WT-UKF) and SVM-2 (WT-UKF) models were established for comparison with SVM-3 (WT-UKF) models. SVM-1 (WT-UKF) and SVM-2 (WT-UKF) models use the WT-UKF algorithm for filtering and detection to obtain feature data sets; however, SVM-1 (WT-UKF) parameters c=10 and g=0.5, respectively, and SVM-2 (WT-UKF) parameters c=10 and g=0.1, respectively. In order to verify the influence of the artificial experience selection of parameters *c* and *g* and IPSO algorithm adaptive selection of parameters *c* and *g* on identity recognition, the IPSO-SVM (WT-UKF) model was established. This model uses the WT-UKF algorithm to filter and detect the feature data set, but parameters *c* and *g* are adaptively selected by the IPSO algorithm. The parameters in the IPSO algorithm are the same as those in Section 3.2.3. The databases used in the five models are the MIT-BIH arrhythmia databases, and the number of subjects (sample categories) participating in the classification together was 30.

As can be seen from Table 5, after the testing set was introduced, with the feature values obtained after processing by the WT-UKF algorithm, the identification accuracy of ECGs of the SVM algorithm was 94.91%, which is 2.23% higher than that of the conventional decision tree algorithm, and the error was reduced by 30%. Under the condition that the SVM model takes the same parameters, with the feature values obtained after processing by this proposed WT-UKF denoising algorithm, the identification accuracy of ECGs was 1.5% higher than that obtained after processing by the conventional WT denoising algorithm, and the error was reduced by 22%. Compared with SVM-1 and SVM-2 under randomly selected parameters, the SVM-3 model with multiple parameter verification to obtain the optimal parameters improved the identification accuracy by 4.83% and 11.25%, respectively, and the error decreased by 48% and 68%, respectively. It can be seen that the artificial experience selection of two parameters in the SVM model had a substantial influence on the identification accuracy. After using the IPSO algorithm instead of the artificial experience to select the parameters, as shown in Figure 10, after eight iterations of IPSO, the fitness, namely, the error function, was reduced to 4.83%, and the recognition accuracy could reach 95.17%. Compared with the SVM model with multiple parameter verification, the error was reduced by 5.11%, and the convergence speed was faster than that of the traditional PSO optimization. Additionally, there was no local optimum.

In order to verify the extent to which the identification accuracy of the method in this paper is affected by the number of sample categories, the maximum individual accuracy (Accuracy${}_{max}$) was defined as the recognition accuracy when the recognition effect was the best among all subjects involved in the recognition, and the minimum individual accuracy (Accuracy${}_{min}$) was defined as the recognition accuracy when the recognition effect was the worst among all subjects involved in the recognition. The total accuracy (Accuracy${}_{total}$) rate was the average recognition accuracy of all the subjects involved in the recognition.

In this paper, the number of subjects (sample categories) participating in the classification was 3, 5, 10, 20, and 30, respectively. The results are shown in Table 6.

As can be seen from Table 6, the corresponding parameter values and identification accuracy of different sample categories were different. As the number of sample categories increased, the comprehensive recognition rate of the method proposed in this paper decreased gradually, but it could still identify the sample categories accurately. When there were 30 sample categories, the parameter value under the maximum individual accuracy obtained by the IPSO algorithm was not a constant value, which verified the inference in Section 4.2.1. When the sample category was 20 and 30, the minimum individual accuracy decreased to 47.5%. It can be concluded from the data that this sample involved the data of subject 106. However, subject 106 also participated in the accuracy recognition when the number of categories was 5 and 10, but the sample with the minimum individual accuracy was subject 101. Therefore, the identification accuracy in this paper was also affected by the number of sample categories. The smaller the number the categories, the higher the accuracy. When the categories increased, the newly added samples affected the recognition rate of other samples, thus reducing the comprehensive recognition rate.

In order to verify the robustness of the method proposed in this paper, three different proportions were used for analysis. The training set accounted for 50%, 70%, and 90%, respectively, and the testing set accounted for 50%, 30%, and 10%, respectively. The classification results are shown in Table 7.

It can be seen from Table 7 that the smaller the proportion of the training set, the lower the accuracy. On the one hand, the reduction in the training set made the model fitting insufficient. On the other hand, the increase in the testing set reduced the robustness of the model. However, the recognition accuracy could still be maintained at a high level. Considering that the high training set will greatly increase the training cost, this paper suggests that 70% of the training set and 30% of the testing set constitute the best proportion of the training set and the testing set, respectively.

In order to verify the practicability of the method in this paper on the ECG data set of subjects without disease, the above operations were repeated using the MIT-BIH normal sinus rhythm database, and the classification results obtained are shown in Table 8.

It can be seen from Table 8 that the method proposed in this paper is also applicable to the classification of normal ECG data. Compared with the results in Table 6, when the number of subjects participating in the identification was 5, the identification accuracy improved by 2.5%, and when the number of subjects increased, it ensured a high identification accuracy.

## 5. Conclusions

The great advantages of ECG signals make it an important part of biometric identification. This paper proposes an ECG signal identification method based on WT-UKF and SVM. First, through simulation experiments, ECG data could be filtered by the WT-UKF algorithm, which could improve the detection accuracy of P waves and T waves while removing noise interference. Then, combined with the obtained eigenvectors, the effectiveness of the SVM multi-classifier in ECG signal identification was verified by artificially setting the SVM’s hyper parameter *g* and penalty parameter *c*. Finally, under the adaptive optimization of the IPSO algorithm, the parameter verification of the conventional SVM algorithm was omitted, and the obtained penalty parameter and hyper parameter could achieve higher recognition accuracy. Additionally, in the MIT-BIH arrhythmia database, when the number of categories changed, the recognition accuracy could reach 100% at the highest level and 95.17% at the total, and it had a better recognition effect. In the MIT-BIH normal sinus rhythm database, when the number of categories changed, the recognition accuracy could reach 100% at the highest level and 96.75% at the total, and it had a better recognition effect.

The next step is to further modify the classification algorithm to reduce the impact of increasing categories.

## Figures and Tables

**Figure 1 sensors-22-01962-f001:**
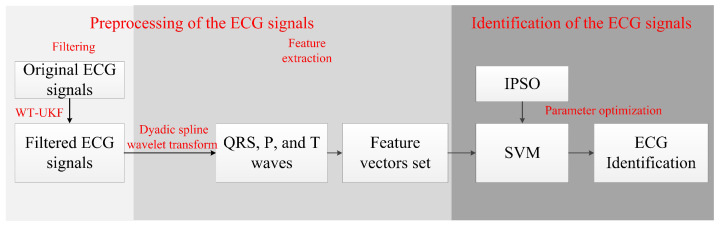
Block diagram of the ECG identification system based on WT-UKF and IPSO-SVM.

**Figure 2 sensors-22-01962-f002:**
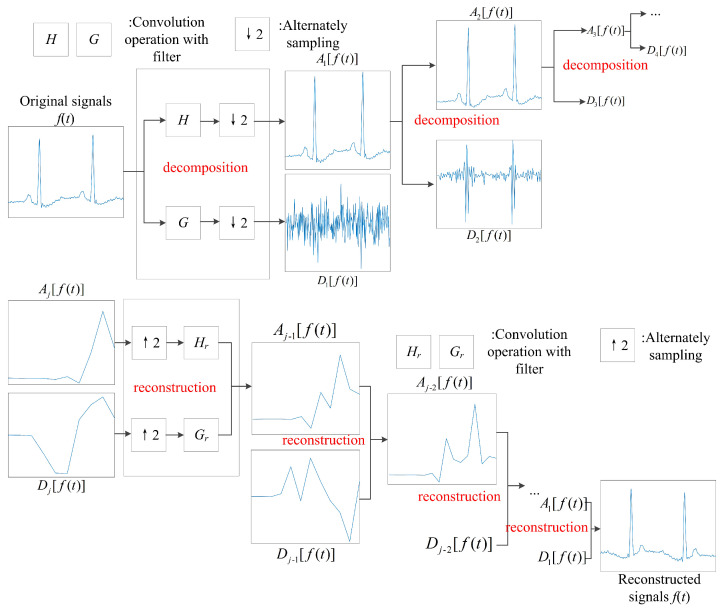
Diagram of wavelet decomposition and reconstruction.

**Figure 3 sensors-22-01962-f003:**
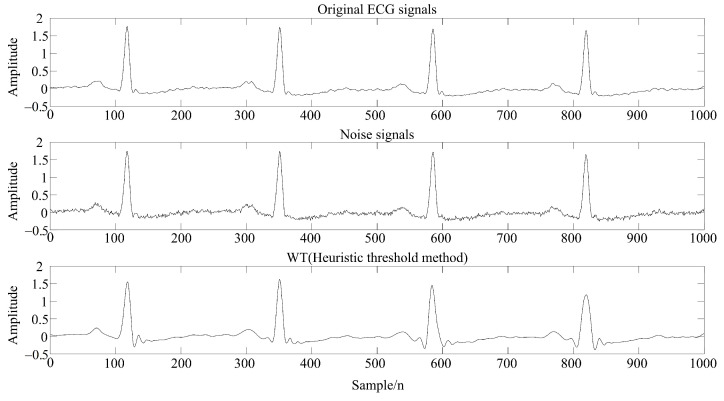
Filtering effect of wavelet threshold denoising (heuristic threshold) of ECG signals.

**Figure 4 sensors-22-01962-f004:**
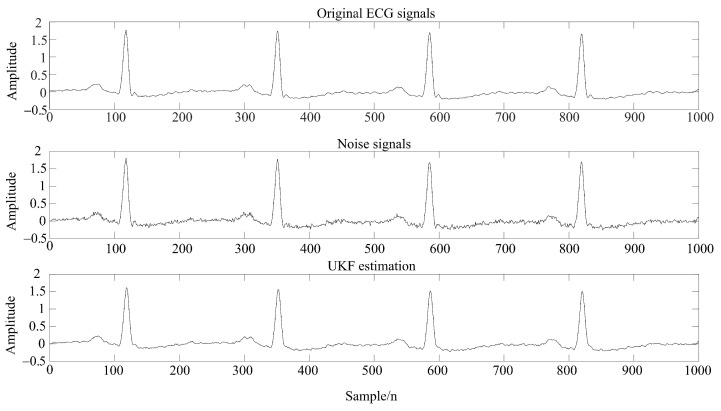
UKF filtering effect of ECG signals.

**Figure 5 sensors-22-01962-f005:**
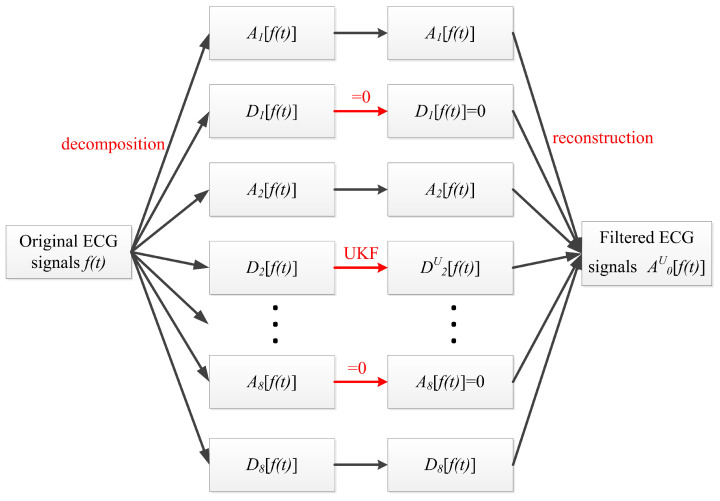
Basic flow chart of WT-UKF algorithm.

**Figure 6 sensors-22-01962-f006:**
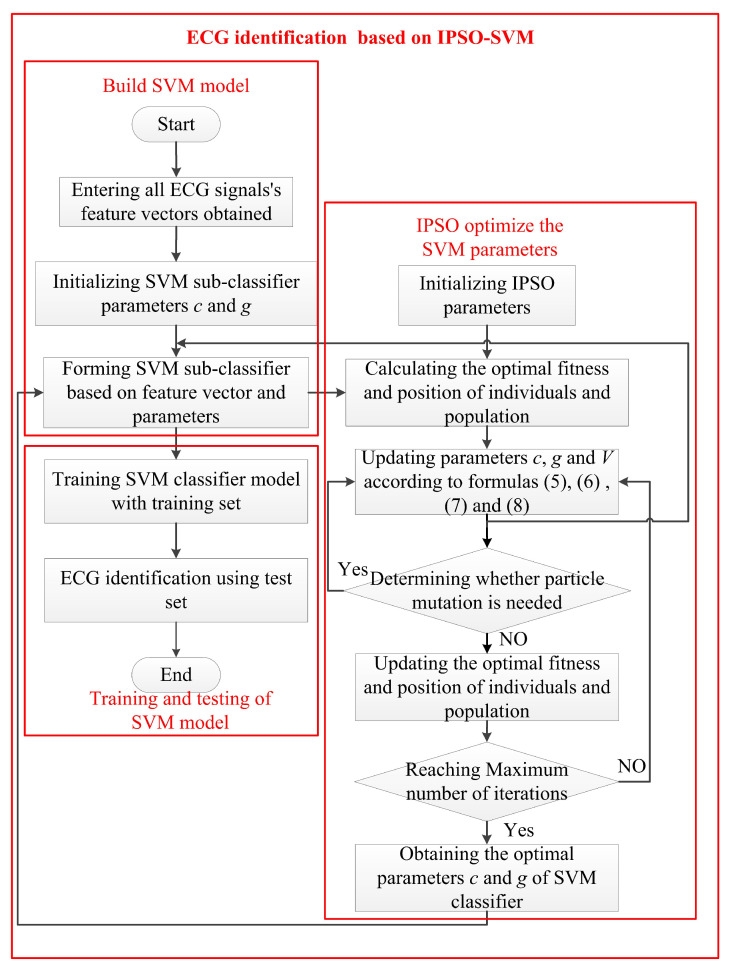
Block diagram of ECG identification based on IPSO-SVM.

**Figure 7 sensors-22-01962-f007:**
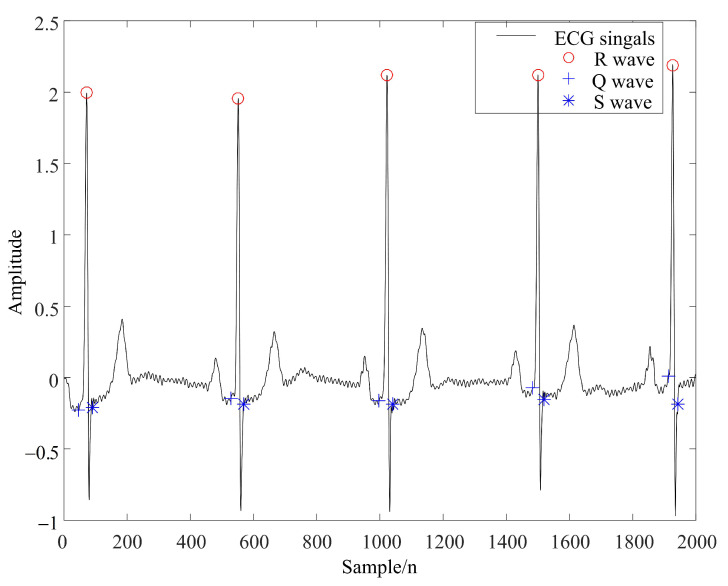
QRS complex extraction results from ECG signals of subject 123 (MIT-BIH arrhythmia database).

**Figure 8 sensors-22-01962-f008:**
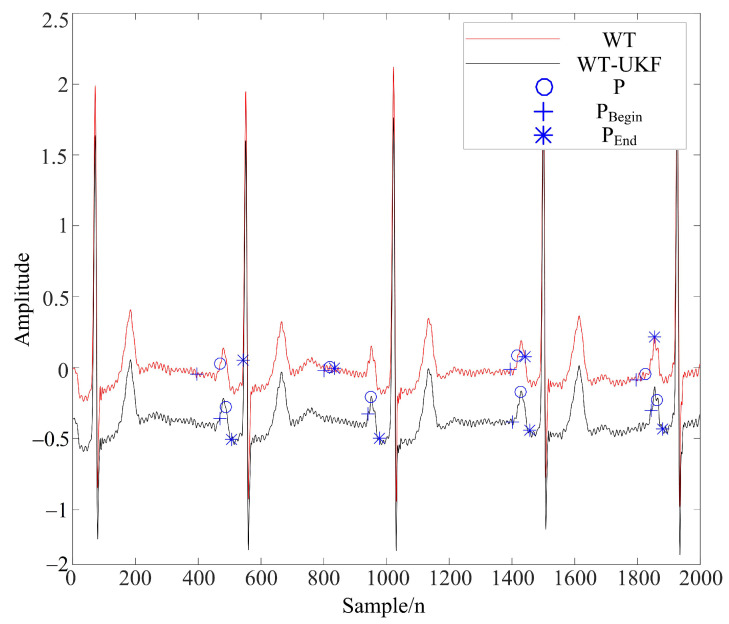
P-wave extraction results from ECG signals of subject 123 (MIT-BIH arrhythmia database).

**Figure 9 sensors-22-01962-f009:**
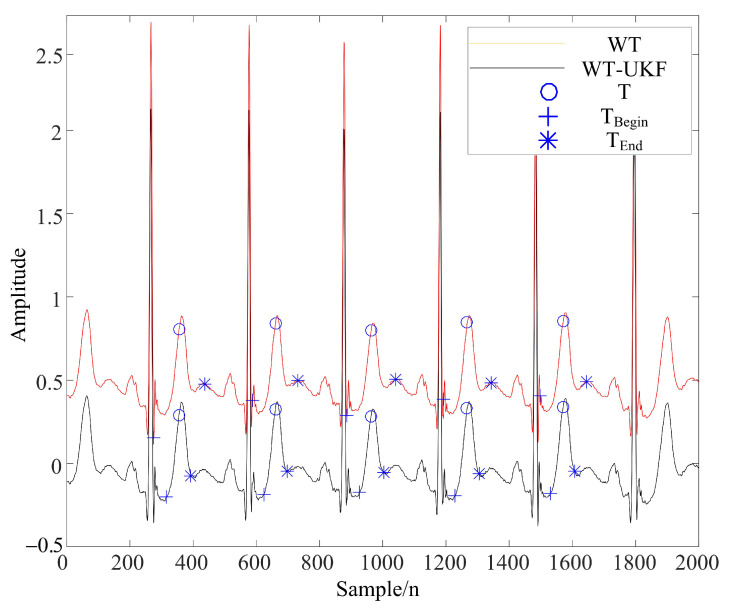
T-wave extraction results from the ECG signals of subject 103(MIT-BIH arrhythmia database).

**Figure 10 sensors-22-01962-f010:**
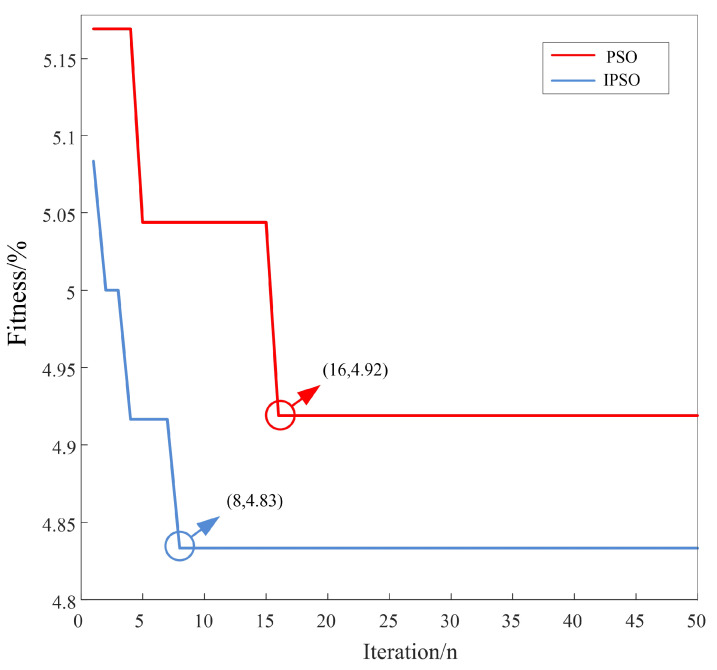
Iteration process diagram of IPSO-SVM algorithm (MIT-BIH arrhythmia database).

**Table 1 sensors-22-01962-t001:** Accuracy estimates of different kernel functions.

Kernel Function	Accuracy Estimate (%)
Linear	93.71
Polynomial	92.64
RBF	95.28

**Table 2 sensors-22-01962-t002:** Comparison of the effects of different denoising methods (MIT-BIH arrhythmia database).

Filtering Algorithm	SNR(db)	RMSE
WT(Heuristic threshold method)	28.12	0.0178
WT(Unbiased estimation adaptive threshold method)	28.11	0.0178
WT-UKF algorithm	32.42	0.0139

**Table 3 sensors-22-01962-t003:** The influence of *c* on accuracy (MIT-BIH arrhythmia database).

*c* Value (When g=1)	Accuracy (%)
2	93.36
2${}^{2}$	94.70
2${}^{4}$	96.33
2${}^{6}$	96.50
2${}^{8}$	96.57
2${}^{10}$	96.73
2${}^{11}$	96.66

**Table 4 sensors-22-01962-t004:** The influence of *g* on accuracy (MIT-BIH arrhythmia database).

*c* Value	*g* Value	Accuracy (%)	Mean Accuracy (%)
$c=2^{8}$	$2^{-4}$	95.70	96.45
	$2^{-3}$	96.20	
	$2^{-1}$	96.73	
	$2^{0}$	96.56	
	$2^{1}$	96.66	
	$2^{2}$	96.40	
	$2^{3}$	95.96	
$c=2^{10}$	$2^{-4}$	95.26	96.47
	$2^{-3}$	96.36	
	$2^{-1}$	96.93	
	$2^{0}$	96.73	
	$2^{1}$	96.70	
	$2^{2}$	96.40	
	$2^{3}$	95.97	
$c=2^{11}$	$2^{-4}$	96.23	96.46
	$2^{-3}$	96.46	
	$2^{-1}$	96.86	
	$2^{0}$	96.66	
	$2^{1}$	96.70	
	$2^{2}$	96.40	
	$2^{3}$	96.96	

**Table 5 sensors-22-01962-t005:** Accuracy comparison of different algorithms (30 sample categories).

Method	Accuracy (%)
Decision tree [40]	92.68
Random Forest [40]	92.68
Bayes [40]	90.24
Logistic [40]	83.54
TCNN-RNN [32]	96.00
CNN [29]	95.20
LSTM [30]	96.45
PNN	94.48
SVM-3 (WT)	93.41
SVM-1 (WT-UKF)	90.08
SVM-2 (WT-UKF)	83.66
SVM-3 (WT-UKF)	94.91
IPSO-SVM (WT-UKF)	95.17

**Table 6 sensors-22-01962-t006:** Accuracy of different numbers of samples (MIT-BIH arrhythmia database).

Categories	*c*	*g*	Accuracy${}_{max}$ (%)	Accuracy${}_{\min}$ (%)	Accuracy${}_{\mathrm{total}}$ (%)
3	928.505	9.413	100.00	100.00	100.00
5	4.000	5.147	100.00	92.50	97.50
10	4.000	4.962	100.00	92.50	97.75
20	1710.000	0.396	100.00	47.50	95.26
30	516.424	0.495	100.00	47.50	95.17
30	575.210	0.423	100.00	47.50	95.17

**Table 7 sensors-22-01962-t007:** Accuracies under different proportions of training sets and testing sets (MIT-BIH arrhythmia database).

Categories	*c*	*g*	Training Set (%)	Testing Set (%)	Accuracy (%)
30	540.505	0.0634	50.00	50.00	92.00
30	575.210	0.4237	70.00	30.00	95.17
30	1423.045	0.1479	90.00	10.00	98.44

**Table 8 sensors-22-01962-t008:** Accuracy of different numbers of sample (MIT-BIH normal sinus rhythm database).

Categories	*c*	*g*	Accuracy${}_{\max}$ (%)	Accuracy${}_{\min}$ (%)	Accuracy${}_{\mathrm{total}}$ (%)
5	2048.000	0.062	100.00	97.50	99.00
10	227.945	0.062	100.00	90.00	96.75

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
