# Peer review of "The Identification of ECG Signals Using WT-UKF and IPSO-SVM"

_sensors, 2022, doi:10.3390/s22051962_

Round 1

Reviewer 1 Report

The paper still needs a careful language revision.

Regarding cross-validation, a reference is missing and there is no need to detail it as much as it is detailed.

It is not clear if the feature vector is composed by the UKF-WT coefficients (some are filtered and other thresholded) directly or if there is another extraction process. The doubt arises since from Fig 1 it is like there is another processing for obtaining the feature vector and Fig 5 indicates that the UKF-WT is used for denoising and therefore this is not clear at all. This is allegedly explained in Section 4.1 but it is not. Is it a set of time samples and amplitudes? Are the differences considered?

The paper presents many broken links, for example as in Section 3.2.1 where there is a reference to section 2.3 as if the hyper-parameter C of the SVM was there defined, but it was not.  

According to Table 4 there is no large gain in using the PSO/IPSO (actually the “I” is about updating the velocity elasticity factor, it is not clear if improved fits). But the points used for the “static” SVM examples in Table 5 are not pinpointed in Table 4 they are rather points of very worst performance, and this seems not be an appropriate claim. It is important to mention how the results for other methods listed in Table 5 were obtained. It is not a problem for the current paper, but all the other methods listed may also presented an hyper-parameters dependent performance, therefore the clear explanation of what is presented is necessary.

It shall be said that the first line of the second paragraph of Section 4.2.2 seem disconnected from what it presents.

An important term to understand the results is “sample category” which is not explained. There are many other terms used without proper clarification as in “the accuracy of the eigenvalue”, for example.

It seems that, once obtained the feature vector of an ECG, then the IPSO is employed for classifying/identifying the person. The authors claim it to be an OVO setup (the person or not the person are the categories). It is not clear how can the IPSO be employed at every new feature vector for on-line optimization in per vector basis.

Reviewer 2 Report

This paper presents a new noise reduction procedure to filter ECG signals preserving local characteristic information and an improved machine learning algorithm combinig SVM and an improved PSO to ensure a higher identification accuracy in ECG identification. The results include a comparison with several classification algorithms and a good analysis of the impact of both the filtering reduction technique and the values of relevant parameters of the classification model. Overall, a relevant improvent is shown when applying these proposals to the MIT-BIH database.

As the paper recognises, the algorithms are tested in a simulated environment, from the point of view of biometric identification. Even when the contributions are valuable, I still think the manuscript tries to focus on biometric identification, which for this case would be a possible application scenario of the denoising, featuring and classification improvements proposed in the work. Thus, some points would need further discussion or revision:

  • The title might be misleading, since would suggest a practical biometric identification context which is not addressed in practice in this paper. A title focused more in the classification problem of ECGs or even in the identification of ECGs would be more appropriate, as are some of the fundamental references.
  • The dataset includes just arrythmia ECG data. This could be not a very general context for biometric identification purposes, although the results would still be very important for identification of arrythmia patients.
  • For each session (patient), the ECG cycles are considered as a time-independent from the point of view of classification, simply separating the first 70% and the last 30% of the 150 cycles per ECG for training and test respectively (by the way, no attention is paid or comment is made on the fact that this selection criteria could have an influence on the results).  So, different variation patterns of ECG features along ECG duration are not considered as relevant to identify a patient, probably because the number of samples in this case would be very short (one per patient).

Some specific considerations about the way some contents are presented:

  • As you indicate in 2.3 (line 173), SVM is essentially a binary classifier. So, it should be better described how are you going to interpret and compose the output for a many class identification.
  • As a consequence, the description of the Min and Max values of Accuracy in Table 6 should  be improved, to ease a better understanding.
  • An explicit mention of the objective function for the noise minimization process in section 3.1 would help a better understanding, in line with the results presented later in Table 2.
  • A bit more of precission in line 210 'After many tests ...' would be valuable.
  • In step 3 of section 3.2.1, you should probably use 'validate and measure' than 'test and measure', since test is later used in a proper way in the final identification experiment.
  • More explicit information on the choose of initial values for c and g in 3.2.3 would be valuable.
  • Some reference and comment to the number of 'genuine' and 'impostor' number of samples (cycles) per subject should be included, at the end of the introductory paragraph of section 4 in order to gain view of the unbalance of the training data.

Some minor formal questions:

  • Filters Hr and Gr mentioned in the description of Figure 2 (lines 156-157) do not appear in the figure (they are shown as h and g).
  • Is 'reflection' used in the correct context in line 182? Should it be 'reduction'?
  • Figure 3 and 4 might not be a good choice to illustrate what is intended. Probably, a representation of the SNR or RMSE as a function of time (for given ECG windows) or similar coud give more information. If the plots are kept, colors should be more adequate (or switch to black and white).
  • In line 279 you refer to 'stop algebra' ... what do you mean? This is not a term which appears as part of the description of IPSO in section 3.2.2.
  • Revise the writing of sentence in line 306. If it is to be understood that the cycle feature extraction from P and T waves and QRS complex is specific of the proposal and does not appear in [37], try a different way to express.
  • Colors in figures 8 and 9 should be chosen differently, to ease a better visual perception.

Reviewer 3 Report

The authors present a study to evaluate a novel method for ECG identification based on multi-scale wavelet transform combined with an unscented Kalman filter algorithm (WT-UKF). The algorithm was verified using signals from the MIT–BIH arrhythmia ECG database.

The paper is well prepared and organized and should be accepted after clarifying the comment below.

Minor Comment:

  1. The authors evaluated the proposed algorithm using the signals from MIT-BIH Arrhythmia Database. The database consists of ambulatory ECG recordings from patients with clinically significant arrhythmias. Thus, this database includes abnormal, pathological signals different from physiological ECG. How does this fact impact the accuracy and identification scores? One could speculate that using abnormal signals will significantly improve the classification score. It can be assumed that the classification of signals with specific manifestations of arrhythmias is simpler than the classification of physiological signals.
